# Growth Potential of *Listeria monocytogenes* on Refrigerated Spinach and Rocket Leaves in Modified Atmosphere Packaging

**DOI:** 10.3390/foods9091211

**Published:** 2020-09-01

**Authors:** Paul Culliney, Achim Schmalenberger

**Affiliations:** Department of Biological Sciences, University of Limerick, V94T9PX Limerick, Ireland; paul.culliney@ul.ie

**Keywords:** *Listeria monocytogenes*, growth potential, ready-to-eat, iceberg lettuce, rocket, spinach, rucola, arugula

## Abstract

Minimally processed ready-to-eat (RTE) vegetables are increasingly consumed for their health benefits. However, they also pose a risk of being ingested with food-borne pathogens. The present study investigated the ability of RTE spinach and rocket to support the growth of *Listeria monocytogenes* as previous studies provided contradicting evidence. Findings were compared to growth on iceberg lettuce that has repeatedly been shown to support growth. Products were inoculated with a three-strain mix of *L. monocytogenes* at 10 and 100 cfu g^−1^ and stored in modified atmosphere (4 kPa O_2_, 8 kPa CO_2_) at 8 °C over 7–9 days. Spinach demonstrated the highest growth potential rate of 2 to 3 log_10_ cfu g^−1^ over a 9-day period with only marginal deterioration in its visual appearance. Growth potential on rocket was around 2 log_10_ cfu g^−1^ over 9 days with considerable deterioration in visual appearance. Growth potential of iceberg lettuce was similar to that of rocket over a 7-day period. Growth curves fitted closely to a linear growth model, indicating none to limited restrictions of growth over the duration of storage. The high growth potentials of *L. monocytogenes* on spinach alongside the limited visual deterioration highlight the potential risks of consuming this raw RTE food product when contaminated.

## 1. Introduction 

The ready-to-eat (RTE) fruit and vegetable industry is a worldwide expanding sector. From 2000 to 2017, global production has increased by approximately 60% for vegetables [1]. Consumption of RTE vegetable salads has also increased within developing countries owing to a change in lifestyle patterns and growth of awareness regarding the positive relationship between human health and intake of RTE vegetables [2]. Indeed, leafy vegetables such as raw rocket and raw baby spinach contain many vitamins, minerals, antioxidants, and phytochemicals [3]. In the European Union, Ireland and Belgium have the highest rate of daily consumption of vegetables (84% of the population; [4]). The health benefits of RTE vegetables have driven consumer lifestyle towards increased consumption of this convenient and healthy type of food in RTE salads and smoothies [5,6]. Within the food industry, demand has increased for variation in terms of taste, color, and shape (in particular, baby sized leafy vegetables) for RTE green leafy vegetables [7].

As RTE vegetables are not at all or only minimally processed from farm to fork, further research is needed to study the risk of consumption of RTE vegetables in relation to foodborne illnesses including listeriosis [8]. Data assessing the occurrences of *Listeria monocytogenes*, the causative agent for the disease listeriosis, in RTE foods from investigations led by the European Union are compiled annually. In the case of fruits and vegetables, 1257 units were tested in 2018 (across 16 Member States) with an overall prevalence of *L. monocytogenes* of 1.8% (up from 0.6% in 2017 across 13 Member States). Additionally, for RTE salads, out of 2583 units, 1.5% of samples were confirmed positive in 2018 for *L. monocytogenes* [9].

Listeriosis can be life threatening, particularly for young, elderly, pregnant women and their unborn baby, and immuno-compromised individuals [10]. *L. monocytogenes* is ubiquitous in nature. It has exceptional physiological abilities to ensure its survival by adapting quickly and easily to harsh divergent physiological conditions [11]. Studies have shown that subjecting *L. monocytogenes* to food-related stresses including low storage (refrigeration) temperatures may induce increased expression levels of the organism’s virulence genes, and thus increase the risk of listeriosis [12].

Storage period and temperature are important factors influencing the growth and survival of *L. monocytogenes* in foods such as RTE salads [13]. However, recent challenge studies also identified that inoculation densities for testing affect the outcomes of challenge studies as lower initial inoculation densities (100 cfu g^−1^) may lead to greater growth potentials during shelf life [14]. Despite the possible underestimation of growth potential, assessing growth potentials at high inoculation densities (i.e., 10^5^ cfu g^−1^) in RTE food remains popular [15]. According to the guidance produced by the European Union Reference laboratory (EURL), if any food product shows growth potential (δ) greater than 0.50 log_10_ cfu g^−1^, it is regarded as being permissive to the growth of *L. monocytogenes* [16,17]. Consequently, changes in the inoculation density or other environmental factors may affect the outcome of challenge studies with the potential to underestimate growth. This could lead to RTE products being falsely categorized as food unable to support the growth of *L. monocytogenes* according to Commission Regulation (EC No 2073/2005) [16]. Up to now, many studies have assessed the prevalence of *L. monocytogenes* on RTE leafy produce [18,19,20], while only few investigated the actual growth potential. In the latter, there have been contradictory findings of growth potential of *L. monocytogenes* on spinach and rocket, of which some reported growth [21], while others did not [22,23]. At the outset of the present study, our hypothesis was that the growth potential of *L. monocytogenes* on spinach and rocket is similar to that of iceberg lettuce. As the growth potential of *L. monocytogenes* on lettuce is well established [14], the consumption of raw spinach and rocket in salads could pose a potential risk for human infection if it is contaminated with low levels of *L. monocytogenes*.

The aim of this study was to determine the growth potential and survival of *L. monocytogenes* during a shelf life study at 8 °C, with initial inoculum densities of 10 and 100 cfu g^−1^ in spinach and rocket (arugula; rucola), and to compare these growth potentials to that of iceberg lettuce. Previously established protocols for testing growth of *L. monocytogenes* on iceberg lettuce were used in this study in order to minimize changes in environmental conditions that could potentially affect growth behavior.

## 2. Materials and Methods 

### 2.1. Preparation of L. monocytogenes for Inoculation Experiments

Three different strains of *L. monocytogenes* from the Teagasc Food Research Centre strain collection (Moorepark, Ireland) were used, 959 (vegetable isolate), 1382 (EUR *Lm* reference strain), and 6179 (food processing plant isolate). For each of the three *L. monocytogenes* strains, 10 mL of tryptone soya broth (TSB, CM0129, Oxoid, Basingstoke, UK) was prepared and placed in 50 mL conical flasks. After autoclaving, single colonies from the previously streaked plates (*Listeria* selective agar, conforming to ALOA) of *L. monocytogenes* culture were transferred into each flask and incubated at 8 °C for 5 days. Spectrophotometry was used to verify the cell density (600 nm) [24]. Dilutions with phosphate buffered saline (PBS, pH 7.3, BR0014, Oxoid) were carried out to mix the three strains at equal cell densities to aim for inoculation at cell densities of either 10 or 100 cfu g^−1^. This was confirmed by enumeration on *Listeria* selective agar, conforming to ALOA (Chromocult^®^ LSA, Merck, Darmstadt, Germany).

### 2.2. Preparation of the Polypropylene Bags and RTE Leafy Vegetables and Subsequent Inoculation and Storage

Oriented polypropylene packaging film (35 μm thick) was used to create storage bags (18 cm × 10 cm) with a permeability to O_2_ of 5.7 nmol m^−2^ s^−1^ kPa^−1^ and to CO_2_ of 19 nmol m^−2^ s^−1^ kPa^−1^ (Amcor Flexibles, 120 Gloucester, UK). Twenty-eight bags were required for each product batch to allow for sampling at day 0, 2, 5, 7, and 9 in quadruplicates and to test for absence of *L. monocytogenes* at day 0 and day end in controls.

On the day of the experiment, the three vegetable products used in this experiment, Iceberg lettuce (Class 1 Spain), spinach (unwashed, origin Italy), and rocket (washed, origin Ireland), were all purchased from the local supplier (Supervalu, Castletroy, Ireland), where they were stored in a refrigerator. The shelf life of all products was at least 7 days at the time of purchase. The whole head of iceberg lettuce was prepared, using disinfected utensils (using 70% isopropanol), by removing the outer two to three layers of the head, and the core and stalk of the lettuce were also discarded. The remaining lettuce was chopped into strips of 1 cm by 3 cm. From this chopped lettuce, 20 g was weighed out and placed into the respective polypropene packaging. Similarly, 20 g of the uncut RTE spinach and rocket leaves was weighed and placed into the polypropene bags. No further processing of the rocket and spinach leaves was carried out (e.g., washing or chlorine dipping). This was repeated for the necessary number of bags prior to inoculation (quadruplicates for five sampling dates, eight bags as non-inoculated controls). Using the previously prepared *L. monocytogenes* dilutions, 100 µL of *L. monocytogenes* suspension (representing 10 or 100 cfu g^−1^ of food product) was distributed uniformly over the 20 g of leafy vegetable product within the polypropene bags (eight control bags were treated with 100 µL sterile PBS [14]). Each packaging containing 20 g sample of vegetable product was then atmospherically treated (8 kPa CO_2_, 4 kPa O_2_, 88 kPa N_2_) using a vacuum packer (Multivac, Dublin, Ireland). The packages were then stored at 8 °C ± 0.5 °C (HR410, Foster Refrigerator, King’s Lynn, UK) for 0–9 days. Storage temperatures were checked daily. Environmental conditions were identical to previous growth studies with iceberg lettuce [14,25].

### 2.3. Sampling of the Leafy RTE Vegetable Packs and Analysis

The specific sampling data points for these experiments were day 0, 2, 5, 7, and 9 (for iceberg lettuce, sampling of day 9 was abandoned owing to a high level of product deterioration). On each of these days, four bags of each product were removed from the storage area. Furthermore, control bags (without *L. monocytogenes* inoculation) were harvested at day 0 and day end, and the absence of *L. monocytogenes* was confirmed on *Listeria* selective agar (conforming to ALOA, see also below), with a detection limit of 1 cfu g^−1^ following methods described previously [25].

Before opening the packs, concentrations of oxygen were determined inside the packs, using a gas analyzer (PBI-Dansensor, PBI Development, Denmark, Model TIA-III LV) with an injection needle to penetrate the packs. Each bag was cut using disinfected utensils (70% iso-propanol), one at a time, directly underneath the heat seal for subsequent sample analysis. Visual appearance was determined on inoculated samples (spinach and rocket), by aseptically removing four leaves from one package for each product at each data point, using disinfected utensils. Images of these leaves and visual markers at each data point were captured using a digital camera. The consumer acceptability was visually assessed for gloss, freshness, and colour uniformity and (given an appearance score) by a sensory panel (postgraduate students, not specifically trained in grading visual appearance of food products [24]) consisting of 10 individuals scoring the products from 1 (mush/very poor condition) to 10 (pristine/excellent condition). A score of 6 was set as the lowest acceptable level for consumption [24]. Images of the samples were all taken in the same artificial light with a visual marker and the same angle, and were then coded and offered randomly to panelists.

Enumeration of *L. monocytogenes* counts was carried out at day 0, 2, 5, 7, and 9. The contents of each package were transferred into separate stomacher bags and homogenized using a stomacher (Seward 400, AGB Scientific, Dublin, Ireland), for 120 s at a high speed (260 rpm), in 20 mL of PBS. Following this, depending on anticipated low cell counts, samples were concentrated (via centrifugation at 4000 g for 240 s) by 10-fold resuspending in 100 µL PBS (10 cfu g^−1^) or 5-fold using 200 µL PBS (100 cfu g^−1^) (detection limit of 1 and 2 cfu g^−1^, respectively, [24]). If necessary, samples were also diluted to achieve a countable number of colonies. Aliquots of 100 µL were then plated on Chromocult *Listeria* selective agar (ALOA) containing *Listeria* selective supplement (both Merck, Darmstadt, Germany). The plates were incubated at 37 °C for 24–48 h. Colony forming units (cfu) on days 0, 2, 5, 7, and 9 were transformed into log_10_ cfu g^−1^, mean values and standard deviations were determined and plotted, areas under the curve were determined [14], and median values were used to calculate growth potentials. Maximum growth rates were calculated as outlined in Appendix B.

### 2.4. Total Bacteria Count

Total bacterial cell counts were repeated (as recommended by EURL) in quadruplicate for spinach and rocket at day 0 and day 9 and for iceberg lettuce at day 0 and day 7. The containments of each package were transferred into separate stomacher bags and homogenised as described above in 20 mL of PBS. Following this, a dilution series was aseptically carried out with PBS and plated on tryptone soy agar (TSA, CM0131, Oxoid). Total bacteria were enumerated after incubation at 37 °C for 48 h.

### 2.5. Product pH and Water Activity

Product pH and water activity were determined (as recommended by EURL) in quadruplicate at days 0, 2, 5, 7, and 9 for each product (day 9 was excluded for iceberg lettuce owing to advanced levels of product deterioration), and average values and standard deviations were reported. For pH measurements on homogenates of each product, a calibrated pH probe (Cole-Parmer, Saint Neots, UK) was used. In order to determine the water activity values, AQUALAB model Series 3TE water activity meter (LabCell Ltd., Four Marks, UK) was used (following the manufacturer’s instructions).

### 2.6. Statistical Analysis

Populations were reported as the means of four replicates and (±) standard deviations and median values were used to calculate growth potentials. The experimental results were tested using SPSS (IBM, Armonk, NY, USA) for homoscedasticity (Leven’s test) and normality (Shapiro–Wilk test). In situations of normality and homoscedasticity, pairwise comparisons (*t*-tests) and analysis of variance (ANOVA) with Tukey post hoc tests were carried out to determine significant differences. When homoscedasticity only was met, then ANOVA with Games–Howell post hoc was carried out. In the case where normality and homoscedasticity were not met, even after data transformation, a Kruskal–Wallis test and manual post hoc was applied in order to identify significant differences (*p* ≤ 0.05 for all tests [14]).

For total bacteria count, the results from each food type were averaged from four replicates. In order to compare results from beginning and end of the experiment and other pairwise comparisons, an independent two-sample equal variance, two-tailed *t*-test was conducted. Significance was determined at *p* ≤ 0.05 [26].

## 3. Results

### 3.1. Comparison of Growth of L. monocytogenes on RTE Leafy Vegetables Iceberg Lettuce, Spinach, and Rocket over 7 Days

The growth of *L. monocytogenes* was supported by all three vegetable products. The growth potentials in all cases exceeded 0.50 log_10_ cfu g^−1^. As cfu for all iceberg lettuce samples was only determined until day 7, all three products were compared based on growth potentials calculated with day 7 being the end day of the shelf life study (Table 1). On the basis of the nine independent 100 cfu g^−1^ experiments carried out on all three products, spinach supported the growth and survival of *L. monocytogenes* on average with the largest growth potential of 2.40 log_10_ cfu g^−1^ (100 cfu g^−1^ δ = 2.16, 2.46, and 2.58 log_10_ cfu g^−1^). This was followed by the average growth potential on iceberg lettuce at 1.86 (100 cfu g^−1^ δ = 1.28, 1.69, and 2.62 log_10_ cfu g^−1^). The average growth potential on rocket was lower at 1.51 (100 cfu g^−1^ δ = 1.08, 1.70, and 1.76 log_10_ cfu g^−1^). The highest growth potential in a single batch was demonstrated by iceberg lettuce (2.62 log_10_ cfu day^−1^). A pairwise comparison of rocket and spinach identified a significant difference (*p* = 0.024, *t*-test). However, a comparison of all three products did not reach significance (*p* > 0.05, ANOVA). Established by the 10 cfu g^−1^ experiments, spinach supported the growth and survival of *L. monocytogenes* on average with the largest growth potential of 2.39 log_10_ cfu g^−1^. However, in contrast to the 100 cfu g^−1^ experiments, this was followed by rocket (1.82 log_10_ cfu g^−1^) and then iceberg lettuce (1.58 log_10_ cfu g^−1^).

In terms of inoculation density for rocket, the 10 cfu g^−1^ inoculum concentration produced a higher growth potential by day 7 than the higher inoculation 100 cfu g^−1^ (see above). For iceberg lettuce and spinach, such a trend was not detected as the growth potentials of 10 and 100 cfu g^−1^ inoculation density overlapped at day 7. Pairwise comparisons conducted on areas under the curve over 7 days identified a significant difference only between spinach and rocket (*p* = 0.02, *t-*test), while a comparison of all three areas under the curve did not reach significance (*p* > 0.05, ANOVA).

Spinach displayed the largest maximum growth rates on average (median) of 0.348 log_10_ cfu day^−1^. Additionally, *L. monocytogenes’* maximum growth rates on lettuce were on average 0.255 log_10_ cfu day^−1^ and on rocket were 0.223 log_10_ cfu day^−1^ (Appendix A).

### 3.2. Comparison of Growth of L. monocytogenes on RTE Leafy Vegetables Spinach and Rocket over 9 Days

Spinach continued to support the growth of *L. monocytogenes* over 9 days (7-day experiments extended by 2 days) with the average largest growth potential of 2.66 (100 cfu g^−1^ δ = 2.36, 2.78 and 2.83 log_10_ cfu g^−1^; Table 2). Rocket supported the growth of *L. monocytogenes* the least, with an average growth potential of 1.83 (100 cfu g^−1^ δ = 1.67, 1.87, and 1.94 log_10_ cfu g^−1^; Table 2). In terms of inoculation density for both spinach and rocket, the lower initial inoculum concentration of 10 cfu g^−1^ leads to greater growth potentials than all 100 cfu g^−1^ experiments by day end (Day 9). At 10 cfu g^−1^ (same conditions), the growth potential (2.90 log_10_ cfu g^−1^) exceeded the highest growth potential in any 100 cfu g^−1^ batch in spinach (2.83 log_10_ cfu g^−1^). (Figure 1A, Table 2). Likewise, when rocket was inoculated with 10 cfu g^−1^, *L. monocytogenes* counts increased by 2.21 log_10_ cfu g^−1^, which was 0.27 log_10_ cfu g^−1^ higher than the highest batch at 100 cfu g^−1^ (Figure 1B, Table 2). A pairwise comparison of the growth potential of rocket and spinach identified a significant difference at day 9 (*p* = 0.007, *t*-test). However, a pairwise comparison conducted on areas under the curve for spinach and rocket, for the duration of 9 days, identified no significant differences (all *p*-values > 0.05).

On the basis of day 9 data plotted to Baranyi and Roberts models (Appendix A), spinach had the highest maximum growth rate on average 0.396 log10 cfu day^−1^. Maximum growth rates for rocket were on average 0.282 log_10_ cfu day^−1^. Day 9 data were also plotted to linear models (Appendix A). There, spinach had the largest maximum growth rate on average at 0.314 log_10_ cfu day^−1^. All maximum growth rates for spinach were higher compared with rocket maximum growth rates, which were on average 0.220 log_10_ cfu day^−1^. Incubation experiments with lettuce were not extended to Day 9 owing to highly advanced levels of deterioration.

### 3.3. Total Bacteria Count

Bacteria counts from spinach, rocket, and iceberg lettuce revealed a log_10_ cfu g^−1^ at day 0 of 6.96, 5.94, and 7.11, respectively. Bacteria counts were also quantified at day 9 (for spinach and rocket) and were 8.86 and 7.97 log_10_ cfu g^−1^, respectively (Appendix A). Iceberg lettuce bacterial counts determined at day 7 were 8.69 log_10_ cfu g^−1^. There were significant increases in the counts of total bacteria for spinach and rocket from day 0 to day 9 and for iceberg lettuce from day 0 to day 7 (*p* < 0.05).

### 3.4. Product pH, Water Activity, and Atmosphere

Spinach’s pH values were highest during the present study. The pH values for spinach were 7.30 (day 0) and 7.25 (day 9), ranging from 6.93 to 7.30 with no trend over 9 days. Rocket’s pH values were 6.55 (day 0) and 6.86 (day 9), ranging from 6.46 to 6.86 also with no trend observed over the course of the 9 days. Iceberg lettuce’s pH values were lowest, at 6.34 (day 0) and 6.36 (day 7), ranging from 6.25 to 6.40 again with no trend demonstrated over 7 days. Water activity values for all three products ranged from 0.970 to 0.996 during this study (Appendix A).

The oxygen concentration in the vegetable packs increased over the first seven days from the initial 4.0–4.2 kPa O_2_ at day 0 to 9.12–11.7 by day 7 (lettuce, spinach, rocket) and, for spinach and rocket, the oxygen concentration stayed at 10.0–10.8 kPa (Appendix A). No significant differences were observed between the three used vegetables.

### 3.5. Visual Appearance of Spinach and Rocket

For spinach (100 cfu g^−1^), visual properties according to the untrained panelists showed a decrease from day 0 to day 9, from a score of above 9 to above 7 (Table 3, Appendix A). This decrease remained above the acceptable limit of 6. In comparison, rocket’s (100 cfu g^−1^) visual appearance analysis decreased from day 0 to day 7 to just above the acceptable range. At day 9, the untrained panelists deemed the visual appearance of rocket to be unacceptable (visual analysis score <6—lowest acceptable commercial score, Table 3).

## 4. Discussion

Previous studies on the growth and survival of *L. monocytogenes* on spinach leaves provided apparently contradictory findings. Lokerse and colleagues [22] demonstrated that, by day 4 of storage/incubation, a relative increase of 0.70 log_10_ cfu g^−1^
*L. monocytogenes* was detectable on spinach at 7 °C. However, by day 5, a significant relative decrease to less than the starting inoculation density at day 0 was detected, which remained until then end of the experiment (day 10) [22]. The authors speculated that antimicrobial compounds present in spinach may cause bacteriostatic activity against *L. monocytogenes* growth. In contrast to the present study, Lokerse and colleagues [22] sealed their spinach in stomacher bags over the duration of the experiment, thus the atmosphere development was likely to be different from the present study, which had an oxygen concentration of around 10 kPa towards the end of the experiment. The same authors also tested the growth potential of *L. monocytogenes* on rocket leaves (rucola). Over the first 9 days of incubation, growth of *L. monocytogenes* on rucola was reported to vary within 0 to 0.9 log_10_ cfu g^−1^, which was close to 0 again by day 9 [22]. Similarly, Söderqvist and colleagues [23] assessed *L. monocytogenes* growth with a starting cell density of 10^3^ cfu g^−1^ at 8 °C on baby spinach (sealed within packages with water vapor and oxygen permeability). There, an increase by 0.30 log_10_ cfu g^−1^ was detected within the first three days, which was followed by a similar decrease by day 7 [23]. These findings are in contrast to the present study, where a continuous growth of *L. monocytogenes* was recorded, albeit at intervals that exceeded 24 h, hence smaller fluctuations between sampling events may have been missed. Nevertheless, Söderqvist and colleagues [23] found substantial growth of *L. monocytogenes* in a mixed-ingredient salad containing baby spinach and chicken, where growth continued to increase over a 7-day period that exceeded 1 log_10_ cfu g^−1^.

Other studies identified growth potentials of *L. monocytogenes* on spinach, which were more similar to the findings from the present study. The validation results from predictive (Bayrani) models that investigated the effect of storage temperature on *L. monocytogenes* on fresh spinach leaves provided reliable estimates [27]. There, results showed growth potentials on spinach, where initial concentrations 2.28 ± 0.47 log_10_ cfu g^−1^ at 8 °C led to maximum population densities of 5.85 ± 0.67 log_10_ cfu g^−1^ over 16 days. With high initial inoculum densities of around 10^5^ cfu g^−1^, growth of *L. monocytogenes* was identified on freshly cut spinach leaves in ambient and modified atmosphere (low O_2_, high CO_2_) over 14 days at 10 °C storage [28]. Interestingly, under ambient atmosphere (filled with atmospheric air), cfu g^−1^ values dropped at day 7, but recovered subsequently to around 10^6^ cfu g^−1^ by day 10. The high starting inoculation density may have played a major role in the more moderate increase in cfu g^−1^ when compared with the study from Omac and colleagues [27]. The findings from Ziegler and colleagues [15] seem to support this hypothesis. They investigated the growth potential of *L. monocytogenes* on rocket (Arugula) at initial inoculation densities of 5.4 log_10_ cfu g^−1^ under environmental conditions similar to the present study. While the authors reported some moderate growth to 5.9 log_10_ cfu g^−1^, this was reported to be not significant.

The determination of growth potentials may have been systematically underestimated in the past and the use of high inoculation densities may have contributed to the contradicting findings of growth of *L. monocytogenes* on spinach and rocket. The present study followed the inoculation density recommendations in ANSES EURL *Lm* technical guidance document for conducting shelf-life studies on *L. monocytogenes* in RTE foods for determining growth potentials in challenge tests [29]. Recently, McManamon and colleagues [14] demonstrated the ability of lower *L. monocytogenes* contamination levels (100 cfu g^−1^) to have higher growth potentials on iceberg lettuce when compared with higher initial densities (10^4^ and 10^5^ cfu g^−1^) at 4 °C and 8 °C. They suggested that, when *L. monocytogenes* reaches higher cell densities (e.g., 10^6^ cfu g^−1^), intra-species competition plays a greater role at limiting growth. This finding could explain why the present study found higher growth potentials on spinach and rocket than the previous studies mentioned above.

Inoculation densities are not the only factor affecting growth of *L. monocytogenes.* According to Beaufort and colleagues [29], pH and water activity are important determinants of *L. monocytogenes* growth; they will not grow when food products have a pH ≤ 4.4 or a water activity value ≤ 0.920 or a combination of pH ≤ 5.0 and water activity ≤ 0.940. In this study, no growth inhibition was expected based on pH and water activity throughout the duration of the experiments for all leafy RTE vegetables tested.

In the present study, spinach had the highest average growth potential, while rocket had a considerably lower growth potential, similar to that of iceberg lettuce. Sant’Ana and colleagues [21] also tested spinach and rocket (among other RTE vegetables) for growth potential of *L. monocytogenes* at 7 and 15 °C. As in the present study, semi-permeable sealed bags were used with a comparable modified atmosphere and the inoculation density was 1000 cfu g^−1^. In their study, rocket had a significantly higher growth potential (1.86 log_10_ cfu g^−1^, 7 °C, day 6) compared with spinach (0.88 log_10_ cfu g^−1^, same conditions). As the differences in results could not be explained by storage conditions, the difference in *L. monocytogenes* strains used and the origin of the produce may have played an important role. In both cases, the leafy vegetables were obtained from local supermarkets, which only revealed the country of origin, and variety or the environmental conditions during cultivation were not revealed. Nevertheless, spinach and rocket came from EU farms in the present study, while Sant’Ana and colleagues [21] received their produce from Brazil. Therefore, one can expect that variety and farming conditions were substantially different.

Growth of RTE vegetables in open fields has been linked to risks of microbial contamination [30]. Research has shown that handling procedures during the harvest greatly influence the presence of food-borne pathogens such as *L. monocytogenes* [31]; thus, environmental abiotic and biotic factors, as well agricultural pre-harvest practices may affect growth of pathogens on leafy vegetables during storage. For example, for products/plants grown in greenhouses or poly-tunnels, controlling the relative humidity (by limiting spells of prolonged high humidity) can serve as an intervention for decreasing *L. monocytogenes* incidences/populations [32]. While the present data clearly support the growth potential of *L. monocytogenes* on spinach and rocket, there is a need to take the pre-harvest environmental conditions as well as the harvest itself into consideration when it comes to identifying growth behavior of *L. monocytogenes* on leafy RTE vegetables in the future.

Natural background microbiota on baby spinach leaves have been reported to potentially affect the growth of *L. monocytogenes* and *Listeria innocua*, although any differences detected were not statistically significant [33]. A general feedback called the ‘Jameson effect’ may be responsible for the limitation of growth that includes *L. monocytogenes* owing to competition for resources when microbial populations are high in numbers. In the present study, total bacteria counts were already high, ranging from 5.94 logs to 7.11 logs cfu g^−1^ for all three tested products, and may have limited growth of *L. monocytogenes* towards day 9. Fitting the growth curve of *L. monocytogenes* on spinach and rocket over 9 days to a linear and sigmoidal function revealed similar relative and absolute measures of fit. This suggests that a slowdown of growth may have just started by day 9. Future experiments with extended storage beyond 9 days might be able to demonstrate this Jameson effect. Bacteria counts of 10^7^ cfu g^−1^ are not uncommon on leafy vegetables. Valentin-Bon and colleagues [34] carried out microbiological counts on both conventional and organic types of spinach (7.7 and 7.2 log_10_ cfu g^−1^) and iceberg lettuce (7.0, 7.3 log_10_ cfu g^−1^), respectively. Likewise, Allende and colleagues [35] found an initial microbial load on baby spinach leaves at 7.2 ± 0.1 log_10_ cfu g^−1^ that increased to 9 logs within 12 days.

Recent work on iceberg lettuce identified a visual degradation of the vegetable that was considered unsuitable for consumption within 5–7 days of storage [24]. Inversely, when the current study identified counts of *L. monocytogenes* on spinach that were higher than on iceberg lettuce, the visual appearance of the spinach decreased less and was still considered to be acceptable by day 9. This is potentially dangerous as, judging from appearance, consumers would likely underestimate the potential risk of high-level contamination with *L. monocytogenes*. In a related study, five panelists (trained in scoring quality attributes) assessed the quality deterioration of commercially packaged baby spinach stored at 8 °C (without contamination) and deemed the product acceptable until day 8 [36]. Fortunately for rocket, its visual quality decreased further than that of spinach in the present study, while growth of *L. monocytogenes* was at unacceptable levels, thus appearance may potentially deter consumers from eating contaminated rocket. Chlorophyll degradation has been identified as the reason for the limited shelf life of rocket [37].

## 5. Conclusions

In conclusion, the present study has confirmed that rocket and especially spinach support the growth of *L. monocytogenes*, with the latter showing very little visual deterioration; therefore, contaminated spinach may pose a serious health risk to consumers. Furthermore, this study identified a range of environmental factors that could explain why many other studies found contradicting evidence of growth of *L. monocytogenes* on rocket and spinach. Indeed, preliminary tests by the authors suggest that rocket cultivated in tunnel or open field influences the natural microbiome of the vegetable, which in turn putatively affects the growth rate of *L. monocytogenes* (data not shown). Therefore, the influence of different varieties of spinach and rocket, soil and climatic conditions, the development of the natural microbiome, and product washing has to be considered in future studies to evaluate the growth potential of *L. monocytogenes* on leafy RTE vegetables in greater detail.

## Figures and Tables

**Figure 1 foods-09-01211-f001:**
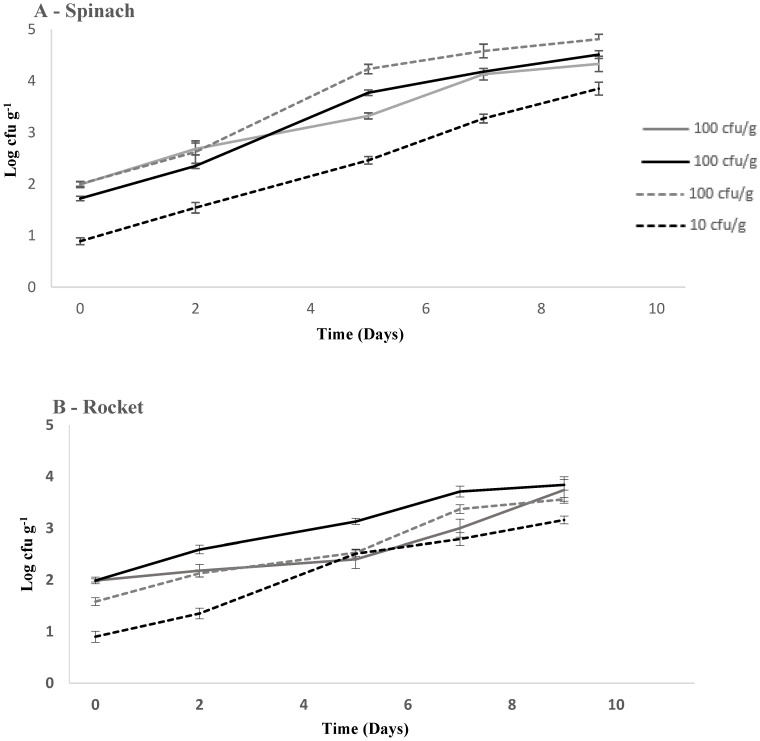
Growth and survival of *L. monocytogenes* in (**A**) spinach, (**B**) rocket, and (**C**) lettuce at 10–100 cfu g^−1^ inoculation densities at 8 °C. (±) error bars indicate standard deviation. Solid black, solid grey, and dashed grey lines represent experiments with 100 cfu g^−1^ starting inoculum density, and dashed black line represents experiment with 10 cfu g^−1^ starting inoculum density.

**Table 1 foods-09-01211-t001:** Growth potentials (based on median values of results from day 0 and day 7) of *L. monocytogenes* in ready-to-eat leafy vegetables.

Product	Batch	Inoculation Density [cfu g^−1^]	Day 0 Median Value (Log_10_ cfu g^−1^)	Day 7 Median Value (Log_10_ cfu g^−1^)	Growth Potential (δ) (Log_10_ cfu g^−1^)
Spinach	1	100	1.98	4.14	2.16
Spinach	2	100	1.73	4.19	2.46
Spinach	3	100	2.02	4.60	2.58
Spinach	4	10	0.88	3.27	2.39
Rocket	1	100	2.00	3.08	1.08
Rocket	2	100	1.61	3.37	1.76
Rocket	3	100	1.99	3.69	1.70
Rocket	4	10	0.92	2.74	1.82
Lettuce	1	100	1.79	3.07	1.28
Lettuce	2	100	1.96	3.65	1.69
Lettuce	3	100	1.34	3.96	2.62
Lettuce	4	10	0.97	2.55	1.58

**Table 2 foods-09-01211-t002:** Growth potentials (based on median values of results from day 0 and day 9) of *L. monocytogenes* in ready-to-eat leafy vegetables.

Product	Batch	Inoculation Density [cfu g^−1^]	Day 0 Median Value (Log_10_ cfu g^−1^)	Day 9 Median Value (Log_10_ cfu g^−1^)	Growth Potential (δ) (Log_10_ cfu g^−1^)
Spinach	1	100	1.98	4.34	2.36
Spinach	2	100	1.73	4.51	2.78
Spinach	3	100	2.02	4.85	2.83
Spinach	4	10	0.88	3.78	2.90
Rocket	1	100	2.00	3.67	1.67
Rocket	2	100	1.61	3.55	1.94
Rocket	3	100	1.99	3.86	1.87
Rocket	4	10	0.92	3.13	2.21

**Table 3 foods-09-01211-t003:** Visual (sensory) analysis of spinach and rocket leaves based on product appearance (i.e., gloss, freshness, and colour uniformity and intensity); average results ± standard deviations. 0 refers to ready-to-eat food products being in poor condition to 10 pristine/excellent condition, with 6 being the lowest acceptable commercial score. ‡ indicates product’s sensory quality is unacceptable at that data point.

	Day 0	Day 2	Day 5	Day 7	Day 9
Spinach 100 cfu g^−1^	9.2 ± 0.8	8.7 ± 0.8	8.2 ± 0.4	7.2 ± 0.6	7.1 ± 0.6
Rocket 100 cfu g^−1^	8.8 ± 0.9	8.7 ± 1.1	6.3 ± 0.5	6.5 ± 0.6	5.6 ± 0.8

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
