# Peer review of "Growth Potential of Listeria monocytogenes on Refrigerated Spinach and Rocket Leaves in Modified Atmosphere Packaging"

_foods, 2020, doi:10.3390/foods9091211_

Round 1
Reviewer 1 Report
The manuscript can be published without further modification.
Author Response
Thank you for the positive evaluation. We are looking forward getting the manuscript published.
Reviewer 2 Report
The manuscript is an interesting paper highlighting the potential growth of Listeria monocytogenes in RTE vegetable salads packed in a modified atmosphere.It represents an important topic for food safety.
It is generally well-written and scientifically valid. The conclusions reached are sufficiently justified. In my opinion the paper can be published without further modifications: I only suggest to revise the alignment style of the References.
Author Response

(The authors gave the same response as above.)

Reviewer 3 Report
The presented study is focused on the evaluation of the growth of L. monocytogenes in chilled green salads (Iceberg lettuce, spinach, rocket leaves) packaged in a modified atmosphere. The experiment is well planned and described. The conclusions reached are sufficiently substantiated.
Line 83-84: I recommend listing the agar manufacturer
Line 86 and other: inconsistent space in front of the unit
Line 91: dot at the end of the title
Line 93: "O2" missing subscript
Line 155: I recommend lowercase letters at water activity
Line 166: enter the T-test without a space
Lines 446, 456, 459, 462: different text alignment style
Reviewer 4 Report
Culliney and Schalenberger in their work have investigated the growth potential of L. monocytogenes on leafy foods. As there are many works showing the growth of L. mono on different type of foods, the study lacks novelty. The effect was assessed only on the mixture of 3 L. mono strains. The number of strains used is very limited. Was the difference in the growth curve between the used strains? As population of L. mono is very heterogenic the growth potential will probably strain-dependent. Therefore more strains should be included in the study. The authors used the inoculum of 10 or 100 cfu/g. For 10 cfu/g inoculum, however, there is only one repetition. More samples should be inoculated with 10 cfu/g. Also the total number of bacteria after each day of sampling should be included.
Round 2
Reviewer 4 Report
The authors’ response letter resloved the reviewers’ doubts. The paper can be accepted in the present form.